# Hemoglobin S and C affect biomechanical membrane properties of *P. falciparum*-infected erythrocytes

Benjamin Fröhlich[1], Julia Jäger[2], Christine Lansche[3], Cecilia P. Sanchez[3], Marek Cyrklaff[3], Bernd Buchholz[4], Serge Theophile Soubeiga[5], Jacque Simpore[5], Hiroaki Ito[6], Ulrich S. Schwarz [2], Michael Lanzer [3] & Motomu Tanaka [1,7]

During intraerythrocytic development, the human malaria parasite *Plasmodium falciparum* alters the mechanical deformability of its host cell. The underpinning biological processes involve gain in parasite mass, changes in the membrane protein compositions, reorganization of the cytoskeletons and its coupling to the plasma membrane, and formation of membrane protrusions, termed knobs. The hemoglobinopathies S and C are known to partially protect carriers from severe malaria, possibly through additional changes in the erythrocyte bio-mechanics, but a detailed quantification of cell mechanics is still missing. Here, we combined flicker spectroscopy and a mathematical model and demonstrated that knob formation strongly suppresses membrane fluctuations by increasing membrane-cytoskeleton coupling. We found that the confinement increased with hemoglobin S but decreases with hemoglobin C in spite of comparable knob densities and diameters. We further found that the membrane bending modulus strongly depends on the hemoglobinopathetic variant, suggesting increased amounts of irreversibly oxidized hemichromes bound to membranes.

[1] Physical Chemistry of Biosystems, Heidelberg University, Im Neuenheimer Feld 253, 69120 Heidelberg, Germany. [2] Institute for Theoretical Physics and BioQuant-Center for Quantitative Biology, Philosophenweg 19, Heidelberg University, 69120 Heidelberg, Germany. [3] Department of Infectious Diseases, Parasitology, Universitätsklinikum Heidelberg, Im Neuenheimer Feld 324, 69120 Heidelberg, Germany. [4] Department of Hematology and Oncology, University Children's Hospital, Medical Faculty Mannheim, 68167 Mannheim, Germany. [5] Biomolecular ResearchCenter Pietro Annigoni, University of Ouagadougou, 01 BP 364 Ouagadougou, Burkina Faso. [6] Department of Mechanical Engineering, Osaka University, Suita, Osaka 565-0871, Japan. [7] Center for Integrative Medicine and Physics, Institute for Advanced Study, Kyoto University, Kyoto 606-8501, Japan. Correspondence and requests for materials should be addressed to U.S.S. (email: schwarz@thphys.uni-heidelberg.de) or to M.L. (email: michael.lanzer@med.uni-heidelberg.de) or to M.T. (email: tanaka@uni-heidelberg.de)

Human red blood cells undergo repeated shape deformations during passage through the vascular system. The ability to temporarily change the cell shape in response to a mechanical stress allows erythrocytes to navigate in capillaries whose diameter is one third of their own size[1]. This capability relies on the fluid nature of the plasma membrane, the high extensibility and, possibly, plasticity of the underlying spectrin/actin network, a homogeneous cytoplasm consisting predominantly of hemoglobin, and a unique cell geometry as defined by a biconcave discoid shape and a high surface area-to-volume ratio[2]. The elasticity of the spectrin/actin network is essential to protect the membrane from rupture and to ensure mechanical stability. Mechanical stability is achieved by the spring-like spectrin tetramers that are connected in hexagonal and pentagonal arrays via actin-containing junctional complexes. The network is anchored to the membrane through the anion exchanger 1 (AE-1; formerly called band 3)-ankyrin and the protein 4.1-glycophorin C macrocomplexes[3,4].

Hereditary red blood cell disorders and infection with the human malaria parasite *Plasmodium falciparum* interfere with this structural organization and, hence, with red blood cell function[5]. For instance, hemoglobin S and C, which deviate from wild-type hemoglobin A by a single amino acid substitution of valine and lysine, respectively, for glutamic acid at position 6 in the ß-globin chain, affect cytoskeletal architecture via a mechanism that involves increased oxidative stress[6,7]. The reactive oxygen species interfere with junctional complex formation and spectrin assembly, among other things[6,7]. As a result, erythrocytes containing homozygously hemoglobin S have a two- to threefold higher shear elasticity compared with wild-type erythrocytes[8,9]. Erythrocytes containing hemoglobin C are also thought to be stiffer, although robust quantitative data are still missing.

The protozoan parasite *P. falciparum* affects red blood cell deformability in various ways. Firstly, the infected erythrocyte swells during the 48 h intraerythrocytic development of the parasite due to the influx of Na+ and accompanying water via parasite-induced channels in the erythrocyte plasma membrane, thereby changing the shape of the red blood cell from biconcave discoidal to almost spherical[10–12]. Secondly, growth of the parasite creates a solid mass that hampers mechanical deformation[13,14]. Thirdly, the parasite remodels the membrane skeleton and, in addition, inserts proteins of its own design into the erythrocyte plasma membrane[15]. For instance, the parasite mines the actin from the junctional complexes in order to establish an actin network required for vesicular trafficking of adhesins and other parasite-encoded virulence factors to the erythrocyte plasma membrane[16]. The adhesins are presented on membrane protrusions, termed knobs, that are anchored to the membrane skeleton[17] and which are thought to contribute to strain hardening[18].

Although there is compelling evidence indicating a loss in deformability in *P. falciparum*-infected erythrocytes[18,19], the detailed roles of the membrane, the cytoskeleton, and their coupling are unclear. Moreover, it is largely unknown how hemoglobinopathies influence the mechanics of parasitized erythrocytes. In fact, it is unlikely that the combination of a hemoglobinopathy and a *P. falciparum* infection contribute in an additive manner to membrane stiffening, since both hemoglobin S and C affect numerous physiological functions of the parasite relevant for membrane mechanics. This includes export of proteins to the erythrocyte surface, host-actin reorganization, and knob formation, with infected hemoglobinopathic erythrocytes displaying reduced anterograde protein trafficking, impaired host actin remodeling and fewer, though grossly enlarged knobs[16,20–23]. Given that the hemoglobinopathies S and C protect

heterozygous carriers form severe malaria[24], a better quantitative understanding of the membrane mechanics might provide novel insights into the underlying protective mechanism, particularly since parasitized HbAS and HbAC erythrocytes display a reduced capacity to cause pathology-mediating cytoadhesive interactions with the microvascular endothelial lining[21,22].

Different biophysical techniques have been applied to quantitatively characterize the mechanical properties of wild-type erythrocytes[25,26]. This includes shear flow experiments[27], micropipette aspiration[28], and optical tweezers[29] probing the mechanical properties of the composite shell of the plasma membrane and the spectrin/actin cytoskeleton. These approaches have also been applied to assess the membrane mechanics of *P. falciparum*-infected erythrocytes[13,19,30]. However, they have merely yielded the effective shear modulus of the composite system, while largely ignoring how the coupling of plasma membrane and the cytoskeleton is changed. A more informative approach to assess the cell mechanics is flicker spectroscopy, which relies on the analysis of thermally or nonthermally activated shape fluctuations[31]. In order to analyze erythrocyte mechanics by flicker spectroscopy[32], it has been suggested that the spectrin/actin network acts as a wall that confines the fluctuations of the plasma membrane[33,34]. This concept allows three mechanical parameters from the fluctuation spectrum to be extracted, namely the bending modulus and the tension of the membrane as well as the membrane confinement parameter characterizing the interaction between the plasma membrane and the spectrin-actin network. By extending this approach into the dynamical domain and measuring the relaxation rates, one can, in addition, extract the cytosolic viscosity[35]. This procedure has been used to show that, in wild-type erythrocytes, the bending modulus and the tension depend strongly on cell shape, but less on oxidation level[36]. Flicker spectroscopy has recently also been applied to erythrocytes decorated with *a P. falciparum*-encoded protein, termed erythrocyte binding antigen 175 that is required for parasite entry[37]. However, it has not yet been applied to *P. falciparum*-infected erythrocytes per se.

Here we have used flicker spectroscopy to quantify changes in red blood cell mechanics during parasite development in wild-type erythrocytes containing hemoglobin A and in erythrocytes containing hemoglobin S or C. We show that the development of the knobs plays an essential role as they increasingly confine the fluctuations of the plasma membrane in the vicinity of the spectrin/actin network. Further, we have combined our experimental data with a mathematical model to better understand the underlying molecular changes. In addition, we show that the membrane bending modulus increased in the hemoglobinopathic erythrocytes, possibly due to the increased level of intrinsic oxidative stress.

## Results

**Dramatic changes in erythrocyte membrane mechanics during *P. falciparum* development**. We first applied flicker spectroscopy to investigate uninfected wild-type erythrocytes (HbAA). The cell contours were extracted from phase contrast videos of flickering erythrocytes. In Fourier space, one expects the mean squared displacement (MSD) of a two-dimensional membrane $<h(q_x, q_y)^2> = k_B T/(\gamma + \sigma q^2 + \kappa q^4)$, where $q = (q_x^2 + q_y^2)^{1/2}$ is the two-dimensional wavevector, $\gamma$ the membrane confinement parameter, $\sigma$ the surface tension, and $\kappa$ the bending modulus. One sees from this equation that fluctuations of small wavelength (large $q$) are suppressed by bending and that fluctuations of large wavelength (small $q$) are suppressed by confinement and tension. Since we recorded the positions of membrane contours at the plane of equator, the equation for the MSD has to be transformed back to

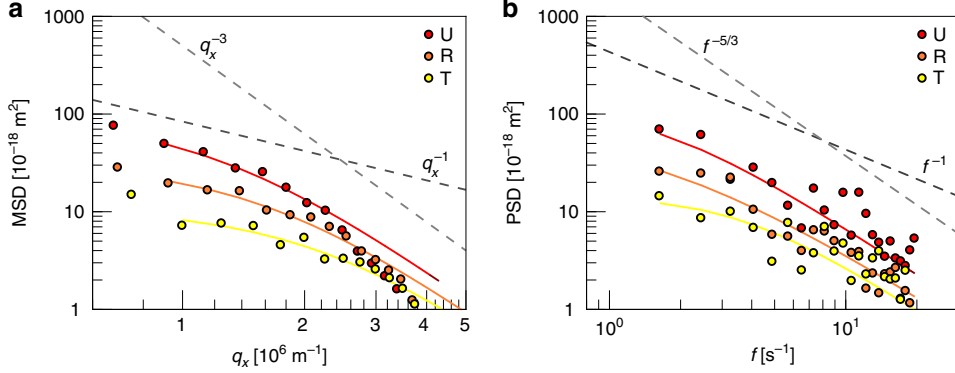

**Fig. 1** Membrane fluctuation spectra obtained from uninfected and *P. falciparum*-infected erythrocytes at the ring and trophozoite stage. **a** Mean squared displacement (MSD) as a function of the wavenumber $q_x$. **b** The corresponding power spectrum density (PSD) as a function of frequency *f*. U (red); uninfected, R (orange); ring phase, and T (yellow); trophozoite. Representative spectra from single cell measurements are shown. The two different power law exponents predicted by the theoretical model for the low and high values of the wavenumber $q_x$ and the frequency *f* respectively, are presented to guide the eye

real space in regard to $q_y$, (Equation 2 in the "Methods" section)[36]. For small and large $q_x$, the MSD should scale with $q_x^{-1}$ and $q_x^{-3}$, respectively. For very small $q_x$, it should level off due to confinement. Figure 1a depicts the measured and fitted fluctuation spectrum for uninfected wild-type erythrocytes (U). The experimental data tend to obey the expected scaling laws, although a clear plateau due to confinement is not apparent. Fitting Equation 2 to the data points yielded values for the membrane bending modulus of $\kappa = (2.7 \pm 0.6) \times 10^{-19}$ N m, the surface tension of $\sigma = (0.7 \pm 0.2) \times 10^{-6}$ N m$^{-1}$, and the membrane confinement of $\gamma = (0.5 \pm 0.2) \times 10^6$ N m$^{-3}$ ($n = 42$). These values are in good agreement with previous determinations[36,38]. In particular, the bending modulus of $\kappa \sim 70 \, k_B T$ is larger than one would expect for bare lipid bilayers (10–20 $k_B T$)[39], indicating a high protein content. The measured surface tension $\sigma$ is well below the lysis tension of a lipid bilayer $\sim 7 \times 10^{-3}$ N m$^{-1}$[40]. The membrane confinement $\gamma$ corresponds to a shear modulus $\mu = \gamma \, r2 = 8 \times 10^{-6}$ N m$^{-1}$, if one takes $r = 4 \, \mu$m as the linear dimension of the erythrocyte projected on a two-dimensional plane. Again this value agrees well with the values reported previously[41].

In Fig. 1b we extended the contour analysis to the temporal domain and plotted the power spectrum density (PSD) against the frequency *f*. By fitting Equation 4 to the data points and taking the formerly determined values for $\kappa$, $\sigma$, and $\gamma$, this analysis allowed us to obtain the apparent red blood cell viscosity of $\eta_{RBC} = (1.2 \pm 0.7) \times 10^{-2}$ N m$^{-2}$ s. This value is consistent with previous determinations by flicker spectroscopy[36], although it exceeds the viscosity of a hemoglobin solution (0.32 mg/mL) by a factor of two[42].

We next investigated wild-type erythrocytes infected with the *P. falciparum* line FCR3 at the ring (observed time window: 10–18 h post invasion) and trophozoite (observed time window: 24–32 h post invasion) stages. These two stages represent different morphological and pathophysiological states of the parasite. Ring-stage-infected erythrocytes circulate through the vascular system, whereas trophozoites are adhesive and sequester in the microvasculature due to the presence of knob-anchored, surface-presented adhesins[43]. As seen in Fig. 1, the intracellular development of *P. falciparum* is associated with a progressive dampening of the shape fluctuation amplitude and the PSD (R and T denote ring and trophozoite stages, respectively). The dampening effect coincided with drastic changes in surface morphology, as indicated by the Gaussian surface roughness (RMS) that changed from 90 ± 34 nm in uninfected erythrocytes to 137 ± 44 nm in rings and to 169 ± 47 nm in trophozoites

(Supplementary Table 1), as determined from phase contrast images. Although the absolute values for uninfected erythrocyte and trophozoites seem larger than what we previously reported using coherent diffraction X-ray imaging[44] due to the difference in instrumental resolution, the increase in the surface roughness is qualitatively in good agreement.

All four mechanical parameters significantly increased as the parasite developed ($p < 0.05$ according to Welch *t*-test) (Fig. 2; Table 1). For instance, the surface tension $\sigma$ increased fourfold to $(3.3 \pm 1.6) \times 10^{-6}$ N m$^{-1}$ in trophozoites, as compared with uninfected erythrocytes (Fig. 2b). Even more pronounced was the increase in the membrane confinement $\gamma$ that rose tenfold to $(5.3 \pm 3.9) \times 10^6$ N m$^{-3}$ (Fig. 2c). In comparison, the change in the bending modulus $\kappa$ was subtler to $(3.4 \pm 1.4) \times 10^{-19}$ N m (Fig. 2a). We further noted a significant, sevenfold rise to $(8.8 \pm 5.2) \times 10^{-2}$ N m$^{-2}$ s in the apparent red blood cell viscosity $\eta_{RBC}$ between uninfected erythrocytes and trophozoites ($p < 0.05$) (Fig. 2d). In summary, the flicker spectroscopy confirmed the expected stiffening during parasite development. However, this effect arose less from a change in bending modulus, but rather from changes in confinement and tension, as evidenced by the dampening of the fluctuations in the low $q_x$ regime.

**Hemoglobin S and C affect membrane mechanics of *P. falciparum*-infected erythrocytes mainly via changes in bending.** We next performed flicker spectroscopy for erythrocytes containing the hemoglobinopathies S and C in their heterologous form with wild-type hemoglobin. The values for $\kappa$, $\sigma$, $\gamma$, and $\eta_{RBC}$ obtained for uninfected and infected erythrocytes were compared with those of wild type (Table 1). Uninfected HbAS and HbAC erythrocytes displayed surface tensions $\sigma$ and membrane confinements $\gamma$ that were significantly higher than those of wild-type erythrocytes ($p < 0.05$) (Table 1). On the other hand, the bending moduli and the apparent red blood cell viscosities $\eta_{RBC}$ of all erythrocyte variants were comparable (Table 1). The comparable level of viscosity (Individual data points presented in Supplementary Fig. 1) seems consistent with recent reports on the mean corpuscular hemoglobin concentrations between uninfected HbAS, HbAC, and HbAA erythrocytes[12]. Infection with *P. falciparum* further altered the membrane properties, leading to gradually increasing mechanical parameters in HbAS and HbAC erythrocytes as the parasite matured, in a manner similar to parasitized wild-type HbAA erythrocytes (Table 1; Fig. 3). Importantly, however, there were clear differences between

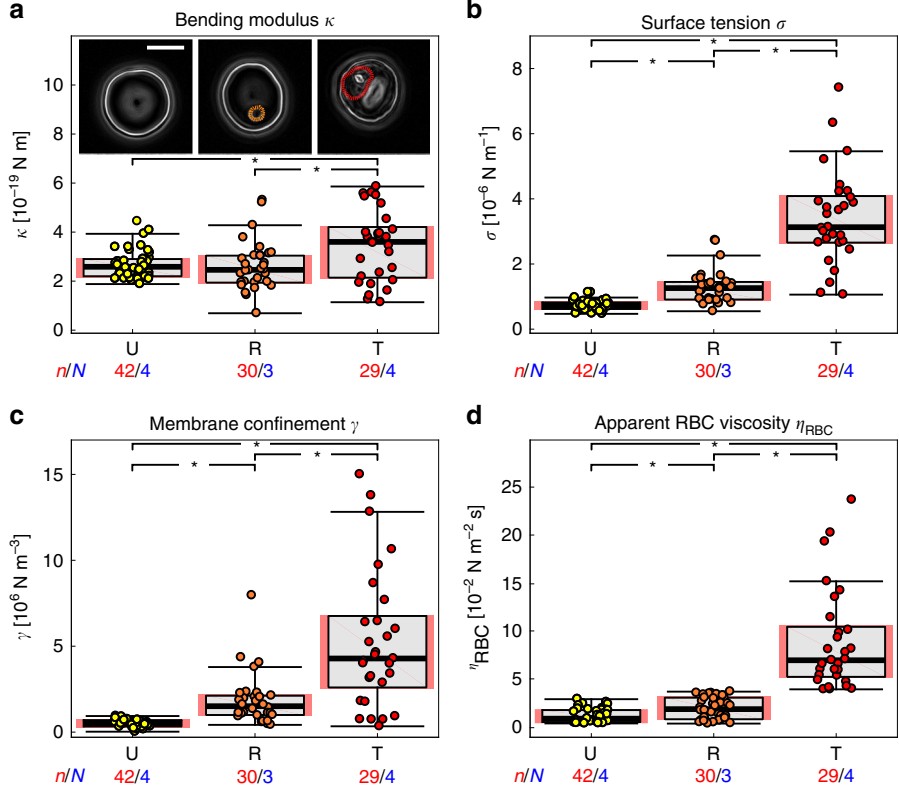

**Fig. 2** Changes in biomechanical parameters in uninfected and *P. falciparum*-infected HbAA erythrocytes. **a** Bending modulus $\kappa$, **b** surface tension $\sigma$, **c** membrane confinement $\gamma$, **d** and apparent red blood cell viscosity $\eta_{RBC}$. U (red); uninfected, R (orange); ring phase, and T (yellow); trophozoite. Individual data points represent a single determination and *n* the total number of data points obtained using blood of *N* different donors. Box plots (gray) were laid over the data points. The 25–75 percentile ranges are highlighted in red and replotted in the following figures. The gradient images of representative cells at the corresponding infection stages are presented as insets of panel **a**. The cytoplasmic area occupied by the parasite is highlighted by the dotted line. *$p <$ 0.05 according to Welch *t*-test. Scale bar, 5 μm

| **Table 1 Biomechanical parameters of uninfected and infected erythrocytes** | | | | |
|---|---|---|---|---|
| | | **Bending modulus $\kappa$ [$10^{-19}$ N m]** | **Surface tension $\sigma$ [$10^{-6}$ N m$^{-1}$]** | **Membrane confinement $\gamma$ [$10^6$ N m$^{-3}$]** | **Apparent RBC viscosity $\eta_{RBC}$ [$10^{-2}$ Nm$^{-2}$ s]** |

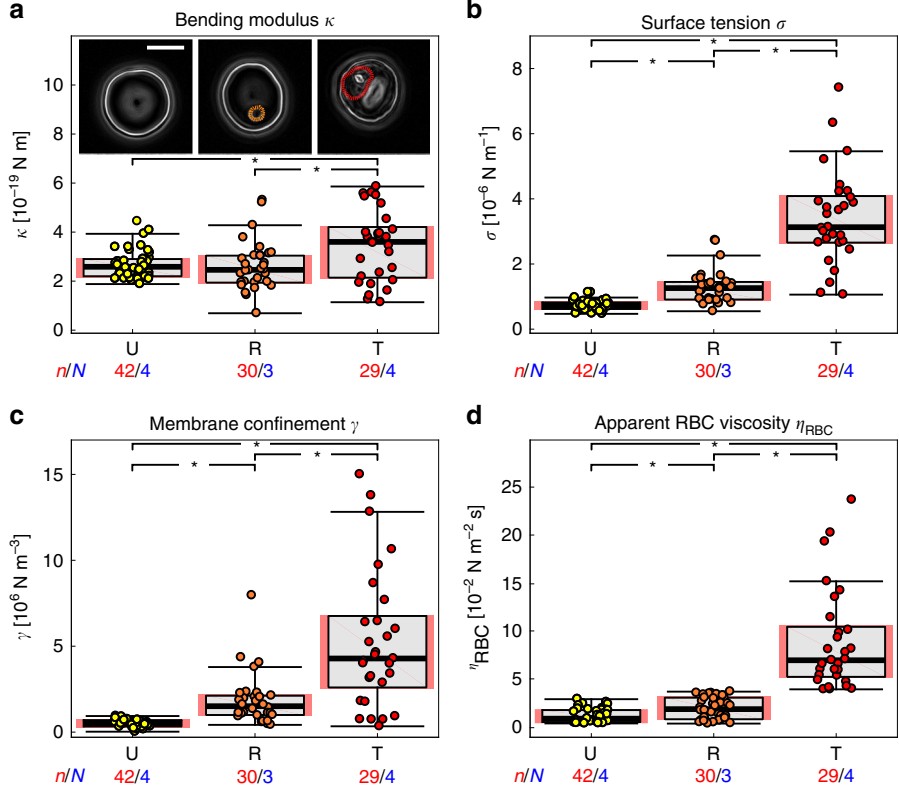

**Fig. 2** Changes in biomechanical parameters in uninfected and *P. falciparum*-infected HbAA erythrocytes. **a** Bending modulus $\kappa$, **b** surface tension $\sigma$, **c** membrane confinement $\gamma$, **d** and apparent red blood cell viscosity $\eta_{RBC}$. U (red); uninfected, R (orange); ring phase, and T (yellow); trophozoite. Individual data points represent a single determination and *n* the total number of data points obtained using blood of *N* different donors. Box plots (gray) were laid over the data points. The 25–75 percentile ranges are highlighted in red and replotted in the following figures. The gradient images of representative cells at the corresponding infection stages are presented as insets of panel **a**. The cytoplasmic area occupied by the parasite is highlighted by the dotted line. *$p <$ 0.05 according to Welch *t*-test. Scale bar, 5 μm

**Table 1 Biomechanical parameters of uninfected and infected erythrocytes**

| | | Bending modulus $\kappa$ [$10^{-19}$ N m] | Surface tension $\sigma$ [$10^{-6}$ N m$^{-1}$] | Membrane confinement $\gamma$ [$10^6$ N m$^{-3}$] | Apparent RBC viscosity $\eta_{RBC}$ [$10^{-2}$ Nm$^{-2}$ s] |
|---|---|---|---|---|---|
| | HbAA | 2.7 ± 0.6 (42) | 0.7 ± 0.2 (42) | 0.5 ± 0.2 (42) | 1.2 ± 0.7 (42) |
| Uninfected | HbAS | 2.8 ± 0.6 (41) | 0.8 ± 0.2 (41)* | 0.6 ± 0.2 (41)* | 0.9 ± 0.6 (39) |
| | HbAC | 2.5 ± 0.8 (36) | 1.1 ± 0.3 (36)*,# | 0.9 ± 0.4 (36)*,# | 1.2 ± 0.6 (36) |
| | HbAA | 2.6 ± 1.0 (30) | 1.3 ± 0.5 (30)$ | 1.9 ± 1.5 (30)$ | 1.9 ± 1.1 (30)$ |
| Ring | HbAS | 2.5 ± 0.8 (25) | 1.4 ± 0.6 (25)$ | 2.3 ± 1.6 (25)$ | 1.9 ± 1.4 (25)$ |
| | HbAC | 3.9 ± 1.4 (25)$,§,& | 2.0 ± 0.6 (25)$,§,& | 2.2 ± 1.0 (25)$,§ | 3.3 ± 1.8 (25)$,§,& |
| | HbAA | 3.4 ± 1.4 (29)$ | 3.3 ± 1.6 (29)$ | 5.3 ± 3.9 (29)$ | 8.8 ± 5.2 (29)$ |
| Trophozoite | HbAS | 5.9 ± 1.2 (39)$,§ | 3.2 ± 1.2 (39)$ | 3.7 ± 2.3 (39)$,§ | 9.2 ± 4.6 (25)$ |
| | HbAC | 5.5 ± 2.2 (40)$,§ | 4.3 ± 1.4 (40)$,§,& | 6.0 ± 3.4 (40)$,& | 9.9 ± 10.5 (25)$ |
| Knobless trophozoite | HbAA | 3.5 ± 1.0 (47)$ | 2.2 ± 0.7 (47)$,§ | 2.2 ± 1.2 (47)$,§ | 9.1 ± 5.4 (34)$ |

The means ± standard deviations are shown for (*n*) independent determination
*$p <$ 0.05 compared with uninfected HbAA erythrocytes; #$p <$ 0.05 compared with uninfected HbAS erythrocytes; $$p <$ 0.05 compared with the corresponding uninfected erythrocyte variant. §$p <$ 0.05 compared with age-matched parasitized HbAA erythrocytes; &$p <$ 0.05 compared with age-matched parasitized HbAS erythrocytes; &$p <$ 0.05 compared with age-matched parasitized HbAS erythrocytes; statistical significance was determined using Welch *t*-test

infected wild-type and hemoglobinopathic erythrocytes as well as between infected HbAS and HbAC erythrocytes. At the trophozoite stage, the bending modulus $\kappa$ significantly increased by a factor of 1.7 and 1.6 in parasitized HbAS and HbAC erythrocytes, respectively, compared with infected wild-type erythrocytes (Table 1; Fig. 3a) ($p <$ 0.05). Importantly, the surface tension $\sigma$ and membrane confinement $\gamma$ showed remarkable differences between HbAS and HbAC erythrocytes at the trophozoite stage ($p <$ 0.05). Both $\sigma$ and $\gamma$ of parasitized HbAC erythrocytes were much larger compared with those of parasitized HbAS erythrocytes (Table 1; Fig. 3c). These results suggest that the two different hemoglobinopathies mainly affect membrane bending through a local mechanism at large $q_x$ (small wavelength), while the difference at small $q_x$ (large wavelength) was more prominent.

**Knobs mainly affect large wavelength fluctuations.** Previous studies have shown that parasitized erythrocytes containing

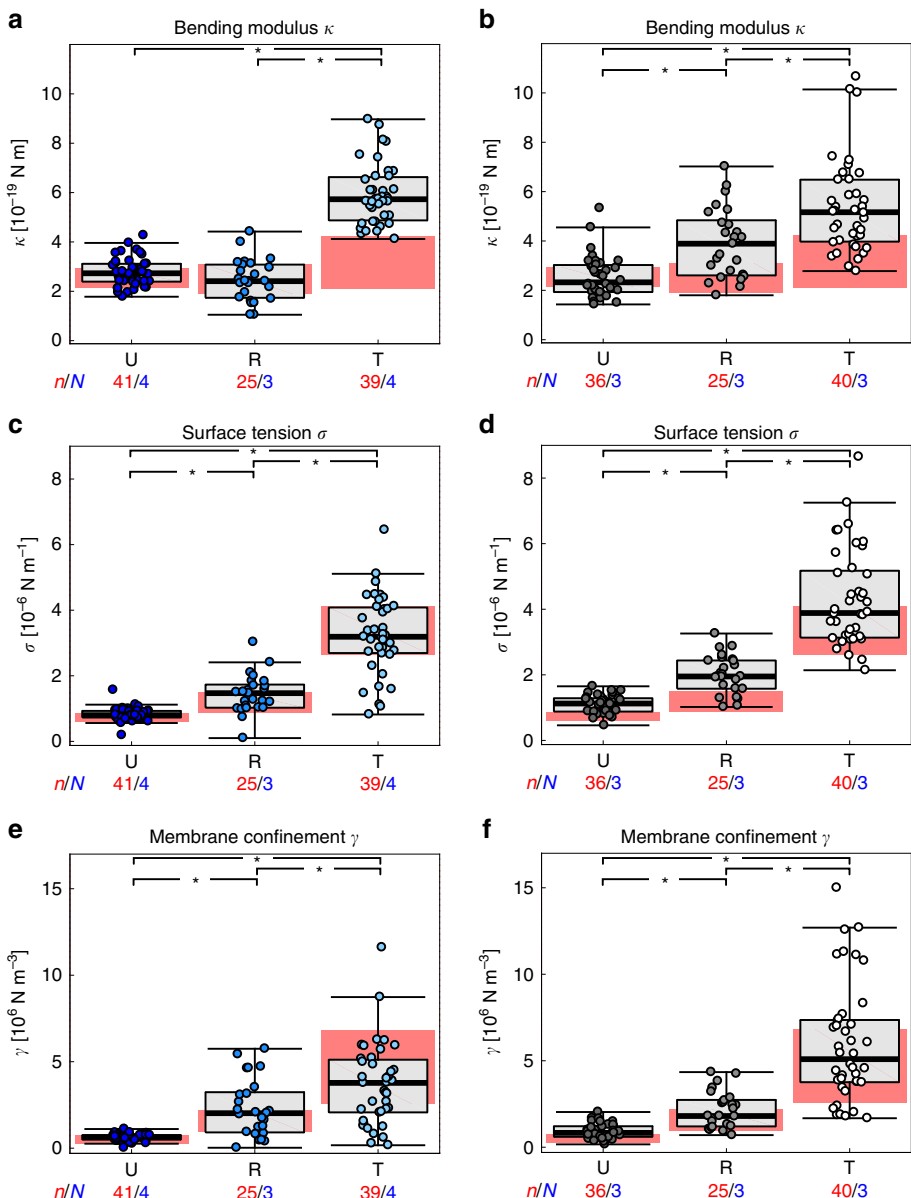

**Fig. 3** Changes in biomechanical parameters in uninfected and *P. falciparum*-infected HbAS and HbAC erythrocytes. **a** Bending modulus $\kappa$ of U (dark blue); uninfected, R (blue); ring phase, and T (light blue); trophozoite HbAS erythrocytes. **b** Bending modulus $\kappa$ of U (dark gray); uninfected, R (gray); ring phase, and T (white); trophozoite HbAC erythrocytes. Corresponding data for the **c** and **d** surface tension $\sigma$ and **e** and **f** membrane confinement $\gamma$. Individual data points represent a single determination and $n$ the total number of data points obtained using blood of $N$ different donors. Box plots (gray) were laid over the data points. The 25–75 percentile ranges of the corresponding HbAA results are highlighted in red for visual comparison. *$p < 0.05$ according to Welch *t*-test

hemoglobin S or C harbor fewer, but bigger, knobs on their surface as compared with parasitized wild-type erythrocytes[16,21,22,45]. We confirmed these findings for the FCR3 strain used in this study, using scanning electron microscopy. Wild-type erythrocytes infected with the FCR3 strain displayed as many as $14 \pm 4$ knobs per $\mu m^2$ at the trophozoite stage, whereas HbAS and HbAC erythrocytes infected with the same parasite line displayed only $3 \pm 2$ knobs per $\mu m^2$ and those few knobs presented were enlarged (Supplementary Fig. 2). To elucidate the impact of the knobs on cell mechanics in general, we investigated a knobless parasite line derived from the FCR3 strain by a chromosomal breakage and healing event within the knob-defining *kahrp* gene, resulting in a truncated nonfunctional *kahrp* gene and the loss of ~100 kbp of DNA from the affected

chromosome 2[46]. Scanning electron microscopy confirmed the knobless phenotype (Fig. 4a). Accordingly, erythrocytes infected with the knobless parasite line displayed a reduced Gaussian roughness of $114 \pm 31$ nm, compared with erythrocytes infected with the knob-forming FCR3 line. Importantly, the knobless parasites revealed a significantly lower membrane confinement $\gamma$ (by a factor of two), as compared with an age-matched control group of the knobby FCR3 strain ($p < 0.05$) (Table 1 and Fig. 4e). Similarly, the surface tension $\sigma$ was distinctly lower (Table 1 and Fig. 4d). On the other hand, the bending modulus $\kappa$ (Table 1 and Fig. 4c) and the apparent viscosity $\eta_{RBC}$ values were similar (Table 1 and Fig. 4f). These findings suggest that the knobs mainly change large wavelength fluctuations due to their anchoring to the membrane skeleton, but that they have

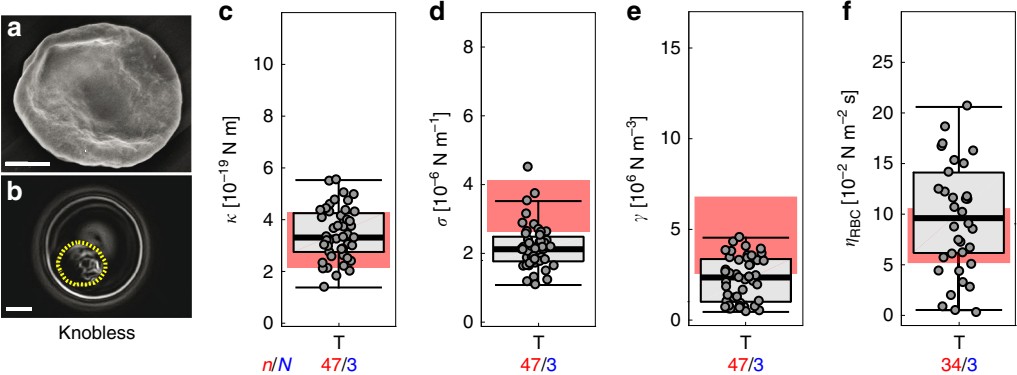

**Fig. 4** Biomechanical properties of HbAA erythrocytes infected with a knobless, FCR3-derived mutant at the trophozoite stage. **a** Representative SEM image and **b** representative gradient map calculated from a phase contrast image. **c** Bending modulus $\kappa$, **d** surface tension $\sigma$ **e** membrane confinement $\gamma$, and **f** apparent red blood cell viscosity $\eta_{RBC}$. Individual data points represent a single determination and $n$ the total number of data points obtained using blood of $N$ different donors. Box plots (gray) were laid over the data points. The 25–75 percentile ranges of the corresponding HbAA results are highlighted in red for visual comparison. *$p < 0.05$ according to Welch $t$-test. Scale bar, 2 µm

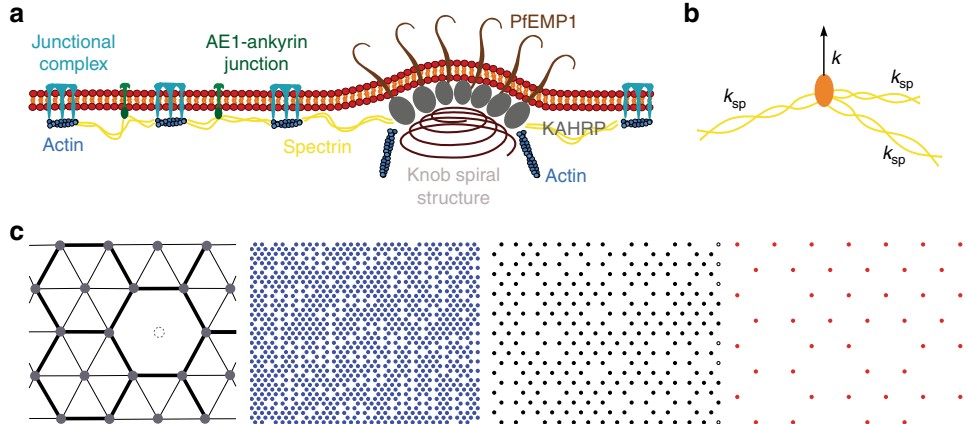

**Fig. 5** Graphic representation of key parameters of the numerical model. **a** Simplified model of the membrane skeleton, including the junctional complexes and AE-1-ankyrin junctions (left, uninfected) and additional anchoring structures via the knob components *Plasmodium falciparum* erythrocyte membrane protein 1 (PfEMP1) and knob-associated histidine rich protein (KAHRP) (right, trophozoite). **b** The structure of a singular anchoring point is depicted, with spectrin filaments indicated in yellow and connections to the lipid bilayer indicated in orange. The whole complex can be modeled by a vertical spring with a spring constant $k$. **c** The model treats the membrane–spectrin connections as springs that are distributed in a hexagonal array. Inhomogeneities are introduced by removing some midpoints of hexagons as indicated in the left panel. The other three arrays show examples of spring distributions for densities of 1096, 257, and 44 springs per µm² (from left to right)

little effect on the small wavelength fluctuations governed by bending.

**Both density and spring constants of knobs have a strong effect on the fluctuation spectrum.** As described above, our experimental data implied that in general knobs strongly affect the membrane confinement $\gamma$ and that this parameter significantly differed between parasitized HbAS and HbAC erythrocytes, in spite of comparable knob densities and knob morphologies (Supplementary Fig. 2) ($p < 0.05$). To identify possible causes for these surprising differences, we used a mathematical model to investigate the effect of the coupling between the lipid bilayer and the underlying cytoskeleton on the fluctuation spectrum. We assumed that the membrane is free to fluctuate only in the regions between the connections to the cytoskeleton, which are progressively remodeled as the parasite develops (Fig. 5a). The model takes into account the density, $\rho$, of membrane–cytoskeleton connectors and their strength as defined by a spring constant $k$ (Fig. 5b). Thus, the model did not distinguish between the molecular nature of the connectors, but focused on their effect on

the membrane fluctuations. The reported spatial heterogeneity in the network was considered by employing a mixture of hexagonal and triangular lattices (Fig. 5c). The fluctuation spectrum was then calculated numerically for a membrane with a given spatial distribution of pinning sites[47,48]. Note that the model calculates the fluctuations of a square membrane patch of ~1 × 1 µm in size with a resolution of 10 nm, and not for an entire red blood cell.

We initially varied the density $\rho$ of the connectors from 44 to 1096 springs µm⁻², while fixing the spring constant $k$ at 13.2 nN m⁻¹. This range of the connector density was chosen for the following reasons. Previous studies have estimated the number of junctional complexes per red blood cell to be around 35,000[3]. Given a surface area of 136 µm² for a red blood cell[49], this converts to 257 junctional complexes per µm². Given that junctional complexes connect between five to eight spectrin filaments, as opposed to the AE-1-ankyrin complexes that bind a single spectrin filament, it is plausible to consider only the connections via the junctional complexes while neglecting the AE-1-ankyrin bridges. Infections with the malaria parasite result in a disassembling of junctional complexes and concomitantly in the formation of knobs, which we took into account by varying

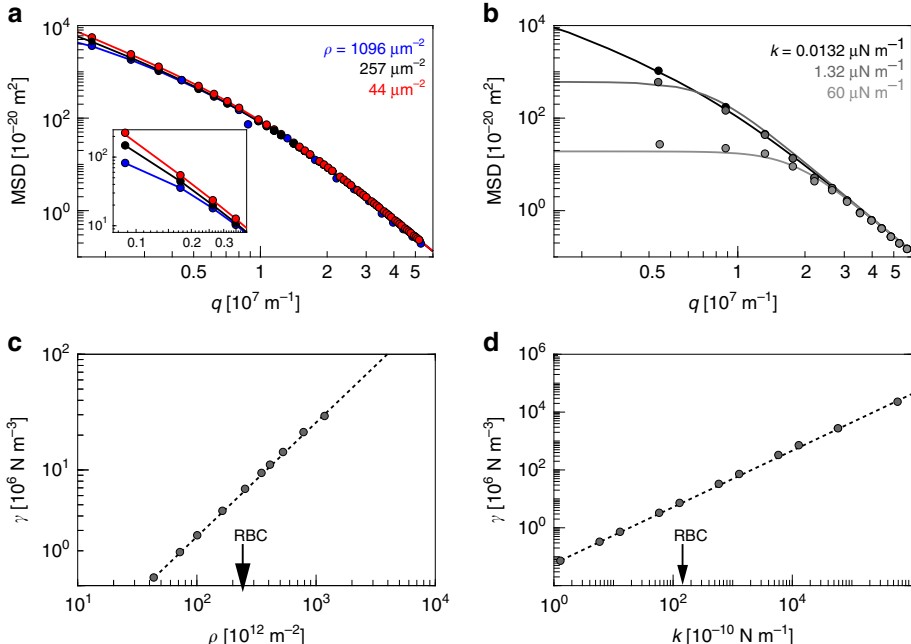

**Fig. 6** Simulated effect of the spring density and the spring constant on the membrane confinement. Simulated mean square displacement as a function of the wavenumber $q$ calculated **a** for different connector densities $\rho$ (see Fig. 5c) and **b** for different spring constants $k$. The discrete data points represent the results of numerical calculations and the continuous lines are the corresponding fit of the continuum theory. Membrane confinement $\gamma$ as a function of **c** the density $\rho$ of connectors **d** the spring constant $k$. The arrows indicate values calculated for uninfected red blood cells

the junctional complex connector density of uninfected wild-type erythrocytes by a factor of five in both directions. A spring constant $k$ of 13.2 nN m$^{-1}$ was chosen given the spring constant of an individual spectrin filament of $k_{sp} = 2$ µN m$^{-1}$ and taking into account that spectrin filaments are arrayed in hexagonal lattices (for further details see methods)[50]. The corresponding fluctuation spectrum calculated from the model revealed that the MSD decreased with increasing connector density in the small wavenumber $q$ regime, indicating a correlation between the connector density $\rho$ and the membrane confinement (Fig. 6a). In the higher $q$ regime, the spectra collapse, consistent with the connectors not affecting the small wavelength fluctuations at equal bending rigidity (Fig. 6a). The global shapes of the calculated spectra seem consistent with that of the experimental ones.

We next varied the spring constant $k$ from $1.32 \times 10^{-10}$ N m$^{-1}$ to $6.0 \times 10^{-5}$ N m$^{-1}$, while keeping the connector density constant at 257 springs per µm$^2$. An increase in the spring constant $k$ strongly affected the MSD, resulting in a major dampening in the low $q$ regime and no effects in the high $q$ regime (Fig. 6b). In order to describe the global effects of changes on connector density $\rho$ and spring constant $k$, we also calculated the effective membrane confinement $\gamma$ corresponding to the microscopic models. The correlations of the membrane confinement $\gamma$ with the connector density $\rho$ and the spring constant $k$ were almost linear (for detailed explanation see supplementary information) over the range of values examined (Fig. 6c, d, respectively).

Finally, we simulated the effect of the knob density and the knob size on the membrane confinement. Knobs were modeled as clusters of springs, with the number of springs per cluster ranging from 13 to 85, corresponding to a projected knob surface area of 3200–39,200 nm$^2$. This range covers experimentally determined knob surface areas obtained from scanning electron microscopy images of HbAA, HbAS, and HbAC erythrocytes (Supplementary Fig. 2). The knob-defining springs with an assumed spring constant $k$ of 26.4 nN m$^{-1}$, were laid over a thinned out

hexagonal lattice in order to capture the situation in an infected erythrocyte. The model assumes a direct correlation between the knob surface area and the number of membrane–cytoskeleton connectors. We found that the membrane confinement directly correlated with the overall number of knob-defining springs and not with their distribution, suggesting that a few but large knobs can compensate for many small knobs (Fig. 7). In summary, these simulations demonstrate that the changes in connector density and spring constants that occur during the progressive establishment of the knob structure recapitulate damping effects on the fluctuation spectrum as observed experimentally.

## Discussion
During intraerythrocytic development, *P. falciparum* modifies the proteinaceous composition of the red blood cell membrane[13]. In addition, the parasite reorganizes the membrane skeleton and it establishes new junctions between the host cell membrane and the remodeled actin/spectrin network[5,16,18]. All of these changes alter the mechanical properties of the host cell membrane, which, in turn, affects the cytoadhesion and clearance mechanisms of red blood cells in the spleen. For example, previous studies demonstrated that the adhesion of HbAS and HbAC trophozoites is weaker compared with that of infected HbAA trophozoites[21,22]. Moreover, using a model spleen system, it has been shown that the rate of splenic clearance is >90% for both HbAA and HbAS trophozoites[51]. This finding suggests that the change in mechanical properties caused by the malaria infection is dominant, irrespective of the hemoglobin types. Using flicker spectroscopy and a mathematical model, our study, quantified these biomechanical changes as a function of parasite development and hemoglobin variants. In particular, it measured defined mechanical parameters, which allowed us to associate them with causative biological processes.

It is well established that the parasite inserts proteins of its own design, such as solute channels and adhesion molecules, into the plasma membrane of the host erythrocyte[52,53]. As a result, the

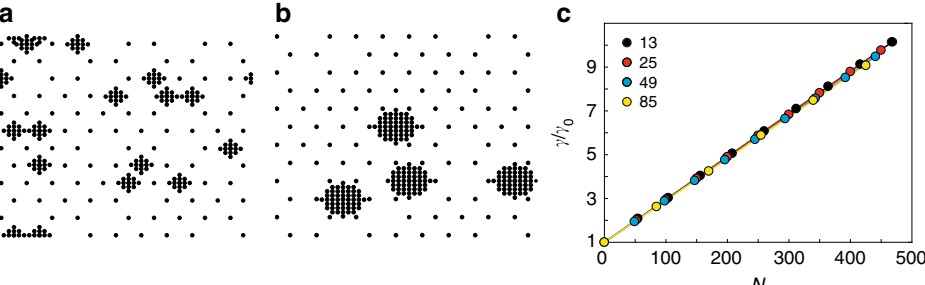

**Fig. 7** Influence of knobs size and density on the membrane confinement. Spring positions of two realizations for a 1 μm × 1 μm membrane patch. Knobs are represented by a clusters of 13 springs and **b** 49 springs. The background represents a membrane with a thinned out anchor density of 102 μm$^{-2}$. **c** The relative membrane confinement $\gamma/\gamma_0$ as normalized to an array without knobs is shown as a function of the cumulative number of knob-specific springs $N$. Each data point represents the mean value obtained from ten individual simulations. Three different knob sizes as defined by spring number were considered. The spring numbers are indicated

plasma membrane could lose flexibility. Indeed, we have measured a clear increase in the bending modulus $\kappa$ (Table 1 and Fig. 2). Interestingly, the increase was more pronounced in infected HbAS and HbAC erythrocytes, as compared with parasitized wild-type red blood cells (Table 1, Fig. 3). We explain this finding by enhanced amounts of membrane cross-linked, irreversibly oxidized haemichromes that are characteristic of hemoglobinopathic erythrocytes due to an increased reactivity of hemoglobin S and C with oxygen[6,7,20]. The metabolic activity of the parasite contributes to oxidative stress and was shown to promote the formation of oxidized haemichromes in HbAS and HbAC erythrocytes[20]. Though there has been no study on the splenic clearance of infected HbAC erythrocytes, the change in the mechanical properties of HbAC trophozoites suggests that the clearance rate should be comparable to those of HbAA and HbAS trophozoites.

Another parameter of global character is the apparent viscosity of the cytosol. Note, that the values we calculated from fluctuation spectra were twofold higher than those of hemoglobin solutions measured by a viscometer[42], although they were consistent with previous determinations using flicker spectroscopy[36]. It is widely accepted that extracting this value from flicker spectroscopy is challenging, often leading to a strong overestimation of the actual values for the cytosolic viscosity[35,54]. Hence, we only regarded the qualitative changes to the apparent red blood cell viscosity during the infection with *P. falciparum*.

Flicker spectroscopy yielded values, particularly for infected erythrocytes at the trophozoite stage, that were unreasonably high despite the fact that the parasite consumes large amounts of hemoglobin during intraerythrocytic development[11]. It has been argued before that the spectrin/actin network rearranges on the timescale of microseconds, while the relaxation time of membranes is on the timescale of milliseconds. Thus, the cytoskeleton should act like a rigid wall in regard to fluid flow between membrane and the spectrin/actin network[34]. Here, we speculate that this assumption becomes progressively less justified as the parasite matures, and that in infected erythrocytes, the erythrocyte cytoplasm can no longer be treated as a uniform viscous fluid and that the flicker-determined values of the apparent viscosity are strongly influenced by the rigid intracellular parasite compartment and, possibly, membrane profiles, such as Maurer's clefts and vesicles, that the parasite establishes into the host cell cytoplasm[14,55].

Our experimental data and the mathematical model revealed an essential influence of the knobs on erythrocyte mechanics, which expresses itself through the membrane confinement revealing dramatic changes during parasite development (Table 1

and Fig. 7). This finding is very plausible. Knobs are large supramolecular structures of ~70 nm in diameter and they are abundant, ~14 ± 4 knobs per μm$^2$ in wild-type erythrocytes infected with the *P. falciparum* FCR3 strain (Supplementary Fig. 2). Knobs are anchored to various components of the cytoskeletal complexes, including spectrin, actin and ankyrin, and they harbor surface exposed adhesion molecules[5,17]. Thus, knobs firmly connect the lipid bilayer with the underlying membrane skeleton and strongly modulate their fluctuations.

We further noted a knob-independent effect on the membrane confinement as revealed by erythrocytes infected with a knobless parasite. The four- to fivefold increase in the membrane confinement in relation to the uninfected control can be explained by the reorganization of actin/spectrin network by the parasite. The parasite mines the actin from the junctional complexes to generate long cortical actin filaments that serve as cables for vesicular transport of parasite-encoded proteins to the erythrocyte surface[16,17]. Consequently, the connections holding spectrin filaments in hexagonal and pentagonal arrays are disassembled, resulting in the spreading of spectrin filaments[56]. An increased end-to-end distance of spectrin filaments leads to a spring hardening effect[56] and, hence, contributes to an enhanced membrane confinement (Fig. 6). The surface tension largely followed the membrane confinement, which is expected since both parameters depend on the coupling between lipid bilayers and cytoskeletons.

Previous studies have shown that parasitized HbAS and HbAC erythrocytes have fewer but larger knobs than wild-type erythrocytes[20–22]. For instance, they possess on average only 3 ± 2 knobs per μm$^2$ and the knobs formed have a basal diameter of ~200 nm, corresponding to an average surface area of ~31,000 nm$^2$ (Supplementary Fig. 2). Simulating the effect of knobs on the membrane confinement revealed that both the knob size and the knob density affected this parameter. Thus, the almost comparable membrane confinements of parasitized wild-type and HbAC erythrocytes at the trophozoite stage could be explained by the compensation of the effect of larger knob size by the lower knob density.

Surprisingly, our measurements revealed a lower membrane confinement for parasitized HbAS erythrocytes than HbAC erythrocytes. This finding is rather unexpected, since both types of infected erythrocytes possess comparable knob densities and knob sizes. Although the underpinning molecular causes are still unclear, our simulations provided a possible explanation by showing that not only the number of connectors but also their strength affected the membrane confinement (Fig. 6b). Thus, the differences in membrane confinement between parasitized HbAS

and HbAC erythrocytes might point towards differences in the anchoring of the knobs to the membrane skeleton. Further work is needed to verify this hypothesis and identify molecular processes that might differently affect the interactions between knob components and the membrane cytoskeleton in red blood cells containing hemoglobin S and C.

The membrane fluctuations investigated here can have both thermal and nonthermal (active) causes[26]. Possible nonthermal causes for membrane fluctuations include phosphorylation events in the spectrin–actin network or in lipid-related molecules like PIP$_2$, known to lead to changes in the shear modulus; the movement of ions through membrane channels and pumps, which transfers momentum to both the membrane and the surrounding fluid; and other membrane-related processes, such as the activity of flippases, floppases and scramblases, or endo- and exocytosis. It is plausible that the parasite changes all of these processes, in particular because it strongly affects ionic currents through the membrane[57]. Recently a four-bead optimal tweezer setup[58] showed that slow fluctuations (below 10 Hz) are mainly active in nature, while fast fluctuations (above 10 Hz) are exclusively thermal in nature. We expect that such an approach can potentially be used to overcome the time resolution of flicker spectroscopy to dissect the active process during parasite development, shedding light on the interplay between the changes in mechanics and activity.

## Methods

**Ethical clearance.** The study was approved by the ethical review boards of Heidelberg University, Germany, Mannheim University, Germany, and the Biomolecular Research Center (CERBA/Labiogene) at the University of Ouagadougou, Burkina Faso. Written informed consent was given by all blood donors.

**Blood phenotyping.** All blood samples were routinely screened by HPLC in the diagnostic hematology laboratory of Heidelberg University Hospital. The phenotype was subsequently confirmed by PCR analysis.

***P. falciparum* culture.** Uninfected HbAA, HbAS, and HbAC erythrocytes were stored in RPMI 1640 medium supplemented with 2 mM L-glutamine, 25 mM Hepes, 100 μM hypoxanthine, and 20 μg mL$^{-1}$ gentamicin at 4 °C until usage. The *P. falciparum* strain FCR3 was kept in continuous culture[59]. Briefly, blood cultures were grown at a hematocrit of 5% and a parasitemia below 5% under controlled atmospheric conditions (3% CO$_2$, 5% O$_2$, and 92% N$_2$, 95% humidity) at 37 °C. Parasites in HbAA, HbAS, and HbAC cultures were synchronized within a time window of 8 h, by sorbitol treatment[60] at time point $t = 0$ h and a repetition at $t \sim 10$ h. Trophozoite stage parasites were enriched by magnetic column isolation[61] at $t \sim 22$ h, starting flicker spectroscopy experiments at $t \sim 24$ h, with an expected parasite age of 24–32 h post invasion. Ring-stage parasites from the following cell cycle were prepared at $t \sim 57$ h and subjected to flicker spectroscopy at $t \sim 58$ h, with an expected parasite age of 10–18 h post invasion. Details on the molecular characterization of the knobless FCR3 mutant are summarized in the supplementary information (Supplementary Fig. 3).

**Flicker spectroscopy.** Cells at different infectious stages (uninfected, ring, and trophozoite stage) were resuspended in RPMI 1640 medium (pH 7.4) supplemented with 1 mg mL$^{-1}$ bovine serum albumin (Sigma-Aldrich) at a hematocrit of ~0.1% and incubated for 30 min at 37 °C in petri dishes with glass bottom. Note that RPMI 1640 contains 2 mg mL$^{-1}$ D-glucose needed to keep a constant ATP level to preserve the activity of the cytoskeleton[62]. The infectious stage was confirmed for each individual cell by microscopic examination (Supplementary Fig. 4). Samples were set inside a temperature-controlled chamber (37 °C) mounted on an Axio Observer Z1 microscope (Zeiss) equipped with a ring aperture, a ×100 oil-immersion objective lens (N.A. = 1.4), and an ORCA-Flash4.0 LT camera (Hamamatsu). Five hundred phase-contrast images of erythrocytes were collected by setting the exposure time and time interval at 25 ms each. Compared with the phase-contrast images, the gradient images give a much better contrast to visualize both parasites surrounded by vacuole membranes and cell membranes (Supplementary Fig. 5).

**Analysis of fluctuation spectra.** The cell rim position $r_{rim}(\theta, t)$ was obtained from gradient images transformed into a polar coordinate ($0 < \theta < 2\pi$) with a step size of $\Delta\theta = 2\pi/256$, where $r_{rim}(\theta, t) = 0$ corresponds to the center of mass of each cell (Supplementary Fig. 6).

The deviation of $r_{rim}(\theta, t)$ from the averaged position was Fourier-transformed into the mean square displacement (MSD) as a function of the wavenumber $q$:

$$\langle u(q)^2 \rangle = \left\langle \left| \frac{2}{N} \sum_{n=0}^{N-1} \{r_{rim}(n\Delta\Theta) - \langle r_{rim}(n\Delta\Theta)\rangle\} e^{\frac{2\pi i (r) qn}{N}} \right|^2 \right\rangle. \quad (1)$$

$< >$ stands for the time average over all frames. The theoretically predicted mean square displacement is described as a function of $q_x$, which is in the continuous wavenumber corresponding to the experimental $q$:

$$\langle u(q_x, y = 0)^2 \rangle = \frac{k_B T}{L} \sqrt{\frac{\kappa}{2(\sigma^2 - 4\kappa\gamma)}} \left| \left[ \frac{1}{\sqrt{2\kappa q_x^2 + \sigma - \sqrt{\sigma^2 - 4\kappa\gamma}}} - \frac{1}{\sqrt{2\kappa q_x^2 + \sigma + \sqrt{\sigma^2 - 4\kappa\gamma}}} \right] \right|. \quad (2)$$

$L$ is the contour length $L = 2\pi \langle r \rangle$ and $\langle r \rangle$ the mean radius of the cell. $\sigma$ is the effective surface tension and $\kappa$ the bending modulus. $\gamma$ is the membrane confinement, which is proportional to the shear modulus $\mu$ of the membrane–cytoskeleton complex. As this equation is not valid for a closed membrane system at small values of $q_x$, the analysis was performed only above $q_x \geq 4$, where the difference between the MSD of a planar and a spherical membrane is $\leq 15\%$[36,38]. Equation 2 predicts that the fluctuation spectrum should exhibit two different power law dependences: the region dominated by the tension ($\sigma \gg \kappa q_x^2$) following $\langle u(q_x, y = 0)^2 \rangle \sim q_x^{-1}$, and the region dominated by the bending modulus ($\sigma \ll \kappa q_x^2$) following $\langle u(q_x, y = 0)^2 \rangle \sim q_x^{-3}$ [36]. To shed light on the dynamics of the system we applied the spherical harmonics approach[35,63]

$$r(\Omega) = \langle r \rangle \left( 1 + \sum_{l,m} u_{lm} Y_{lm}(\Omega) \right), \quad (3)$$

where $\Omega$ is the solid angle, $\langle r \rangle$ the mean radius of the cell, $u_{lm}$ the amplitude corresponding to the $l$, $m$ mode, and $Y_{lm}$ the spherical harmonics. The theoretical PSD as a function of the angular frequency $\omega$ is given by:

$$PSD(\omega) = \langle r \rangle^2 \sum_{l=2}^{l_{max}} \langle |u_{lm}|^2 \rangle \frac{\omega_l}{\omega_l^2 + \omega^2} \frac{2l+1}{2\pi}. \quad (4)$$

$\langle |u_{lm}|^2 \rangle$ is the mean squared fluctuation amplitude for the $l,m$ mode following:

$$\langle |u_{lm}|^2 \rangle = \frac{k_B T}{\kappa(l+2)(l-1)l(l+1) + \sigma\langle r \rangle^2 (l+2)(l-1) + \gamma\langle r \rangle^4}, \quad (5)$$

and $\omega_l$ is the associated characteristic decay frequency:

$$\omega_l = \frac{\kappa(l+2)(l-1)l(l+1) + \sigma\langle r \rangle^2(l+2)(l-1) + \gamma\langle r \rangle^4}{(\eta_{buffer} + \eta_{RBC})/2\langle r \rangle^3 Z(l)}. \quad (6)$$

$Z(l)$ is defined as:

$$Z_l = \frac{(2l+1)(2l^2 + 2l - 1)}{l(l+1)} \quad (7)$$

Equation 4 predicts that the PSD for low frequencies is dominated by the membrane tension $\sigma$, following a power law $f^{-1}$, while the PSD for high frequencies is dominated by the bending modulus $\kappa$ following $f^{-5/3}$ [35]. The apparent RBC viscosity $\eta_{RBC}$ was obtained from equation 4 using the three principal parameters bending modulus $\kappa$, surface tension $\sigma$, and cytoskeletal membrane confinement $\gamma$ obtained from equation 2, and using a buffer viscosity of $0.8 \times 10^{-3}$ N m$^{-2}$ s[64]. The data were analyzed using self-written macros in MatLab.

**Numerical model.** In order to better understand the experimentally observed differences in mechanical parameters during the parasite maturation, we employ numerical calculations connecting the membrane continuum theory with structural changes of the RBC spectrin network on the microscopic scale. Specifically, the confinement parameter $\gamma$ can be well explained with discrete connections between the phospholipid bilayer and the spectrin network where the anchor points are modeled as springs with spring constant $k$ in the vertical direction (see Fig. 5a). There are two major contributions that set the strength of the spring constant. Firstly, the anchoring complex itself behaves spring-like and secondly, during vertical displacements the spectrin network is stretched out of its plane which results in a force in $z$-direction as can be seen in Fig. 5b. Since the parasite alters the spectrin network while it grows, we consider this as the main contribution.

The red blood cell membrane is described by a Hamiltonian consisting of two distinct contributions similarly to the approaches by Lin and Brown[47,48]. The elastic contribution is given by the Canham–Helfrich Hamiltonian, in Monge gauge, which takes into account the bending of the bilayer $\kappa$ and a surface tension $\sigma$:

$$H_{elastic} = \int_0^{L_x} dx \int_0^{L_y} dy \left( \frac{\kappa}{2} [\nabla^2 h(\vec{r})]^2 + \frac{\sigma}{2} [\nabla h(\vec{r})]^2 \right), \quad (8)$$

where $h(\vec{r})$ is the normal displacement ($z$-direction) of the membrane from its equilibrium position in the $xy$-plane. The expression needs to be integrated over the whole membrane patch with the dimensions $L_x$ and $L_y$. The contribution from

the springs consists of a sum over the discrete attachment sites ($\alpha = 1, ..., N$):

$$H_{spring} = \sum_{\alpha=1}^{N} \frac{k_\alpha}{2} h^2(\vec{r}_\alpha), \qquad (9)$$

where $k_\alpha$ is the spring constant at position $r_\alpha$. For the simplicity the spring constants $k_\alpha$ are set to be equal to $k$ for a given realization unless stated otherwise. After Fourier transforming the above equations, an expression for the mean squared amplitudes can be derived using the equipartition theorem. Details about the calculation can be found the work by Lin and Brown[65]. The numerical treatment which is necessary because of the discrete attachment sites, limits the size of the membrane patch. Here we calculate the fluctuations for square patches of 1 μm × 1 μm size with a resolution of 10 nm. By fitting the corresponding formula[34] to the MSD, the values of the relevant parameters can be extracted similarly to the analysis of the experimental data.

The springs, which mainly present junctional complexes, are arranged in a hexagonal array if not otherwise stated (Fig. 5c). This mimics the hexagonal structure of the red blood cell cytoskeleton. However, since the cytoskeleton has been found to have irregularities in form of large voids[66], we take away 49% of the hexagonal midpoints in each configuration (Fig. 5c).

**Estimation of effective spectrin spring constant**. For the numerical calculations we are interested in an effective spring constant for out of plane motion. In order to estimate a realistic value, we neglect the strain hardening property of the spectrin filaments and model it as a simple spring. An individual spectrin filament has been estimated to have a spring constant of $\sim k_{sp} = 2 \times 10^{-6}$ N m$^{-1}$ when extended along its contour length[50] and we assume that an average of six filaments are attached to a junction[4]. By a simple force balance equation, the following formula can be derived:

$$k \approx 6 \, k_{sp} \frac{z^2}{2a^2} \approx 1.32 \times 10^{-8} N \, m^{-1}, \qquad (10)$$

where $\alpha$ is the rest length of the spectrin filament and $z$ the typical displacement orthogonal to the membrane ($z/\alpha \sim 0.0465$).

**Statistics and reproducibility**. The presented data were obtained during at least four different experimental sessions, using blood from at least three different donors of each erythrocyte variant. Statistical significance was analyzed using the Welch $t$-test and the one-sided Anova test, within the MatLab software package.

Please note, that values for all four mechanical parameters obtained for erythrocytes from different donors exhibited no significant differences according to one-sided Anova test ($p < 0.05$) for HbAA, HbAS, and HbAC for the uninfected, ring, and trophozoite state.

**Reporting summary**. Further information on research design is available in the Nature Research Reporting Summary linked to this article.

## Data availability

The authors declare that the main data supporting the findings in this study are available within this article (and the supplementary information). Data of individual determinations of the mechanical parameters presented in Figs. 2, 3, and 4 and the corresponding statistical analyses are available at https://doi.org/10.17632/bfn8p44vcr.1.

## Code availability

The utilized computer code is deposited at https://doi.org/10.5281/zenodo.2657921.

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

## Acknowledgements

This work was supported by the Deutsche Forschungsgemeinschaft through the Collaborative Research Center SFB 1129. U.S.S., M.T., and M.L. are members of the cluster of excellence CellNetworks. C.L. and J.J. were supported by the HBIGS and HGS MathComp graduate schools, respectively, at Heidelberg University. M.T. thanks Nakatani Foundation for supports.

## Author contributions

M.T., U.S.S and M.L. designed the study; B.F., J.J., C.L., C.P.S. and M.C. performed the experiments; B.F., J.J., C.L., H.I., U.S.S., M.L. and M.T. analyzed the data. B.B., S.T.S, J.S. characterized the blood samples and provided the samples; B.F., J.J., U.S.S., M.L. and M.T. wrote the paper. All authors participated in discussion and paper editing.

## Additional information

**Competing interests:** The authors declare no competing interests.

