## [Peer Review File · Communications Biology]

Reviewers' comments:

Reviewer #1 (Remarks to the Author):

The manuscript by Froehlich et al. describes the measurements of biomechanical properties of the plasma membrane of *P. falciparum* infected red blood cells. The changes in physical parameter in the host membrane are significant because they influence pathogenicity and immunoevasion in the host. To measure the properties of the membrane the authors employed flicker spectroscopy – a technique that has not been fully explored for this particular purpose before. The advantage of the technique is that the influence of the plasma membrane and the influence of the coupling to the cytoskeleton on the overall shear modulus can be separated. Based on their measurements and values available in the literature the authors then developed a mathematical model that predicts the influence of the coupling of the membrane to the underlying cytoskeleton on the fluctuation spectrum. In order to tease out the effect of different host cell and parasite modifications the authors compared membrane properties of RBC of the haemoglobin A (WT), C and S phenotype in the infected and uninfected status as well as infected with parasites that do not produce any knobs (parasite induced RBC membrane modifications).

The major findings of the paper are that *P. falciparum* infected HbAS red blood cells display a higher membrane confinement and infected HbAC red blood cells display a lower membrane confinement in comparison to infected HbAA red blood cells. At the same time this has no influence on the number and diameter of the knobs. The authors found that the different haemoglobin variants influence the bending modulus of the membrane differently and they suggest that this is due to the different oxidising states of haemichromes.

These findings are novel and provide new insights into the biomechanics of mutant and *P. falciparum* infected red blood cells, however how useful they will be for a wider audience is difficult to judge. The provided measurements and the mathematical model will provide the foundation of further studies into the effect of membrane modifications on the biomechanics of red blood cells.

Overall, the data presentation is clear and convincing. However, I suggest the authors also consider the following:

- The knobless cell line is not described in the literature reference (46); it outlines the mechanism that results in the absence of knobs (WT FCR3 was used as the knob forming variant) and prolonged culture will result in FCR3 with truncated KAHRP genes, which I assume was done. However, relevant here is that the truncation of KAHRP might result in the absence of visible knobs, but not necessarily in the absence of the complete knob structure (Rug et al., Blood 2006). This needs to be taken into consideration when discussing the results.

- Will the difference observed in the different Hb variations likely result in differences in splenic clearance and cytoadherence of infected red blood cells?

- The influence of lipid composition on membranes has not been considered.

- The text and figures could be more integrated:

The manuscript is based on the comparisons of the different Hb mutations/parasite strains. This is in contrast to the organisation of the figures, which show different measurements of one mutation/strain. It would be more helpful to put all the measurements of the bending modulus in one figure, followed by a comparison of the surface tension, etc.

On occasions it might be difficult for non-experts to follow the flow of the arguments. This can be addressed relatively easily by for example introducing the word "confinement" as "membrane confinement" in the abstract or linking conclusions more closely to the data or preceding text (e.g. line 190/line 386)

- Line 163-165: what is the explanation that the measured values exceed the viscosity of haemoglobin solution by a factor of two?

- Line 315-317: insertion of proteins in plasma membranes does not necessarily result in the loss of membrane flexibility.

- Line 343: the word "extruding" for the Maurer's cleft genesis seemed to be unfortunate
- Line 371: Is there further support for the correlation that larger knob sizes influence the harmonic confinement?

Reviewer #2 (Remarks to the Author):

This manuscript applies flicker microscopy to understand the biophysical changes that take place in erythrocytes infected with *Plasmodium falciparum* parasites. The parameters generated in infected and uninfected erythrocytes are compared between erythrocytes obtained from individuals who express the Haemoglobin beta variants HbS and HbC, which in heterozygotes provide protection against severe malaria. The experimentally derived values are then compared with values derived from a novel mathematical modelling approach.

Flicker microscopy has only recently been applied to understanding biophysical properties of erythrocytes, but not to my knowledge previously to *P. falciparum* infected erythrocytes. Previous comparable studies (which are referenced) have either only looked at uninfected erythrocytes, or uninfected erythrocytes exposed to *P. falciparum* adhesins. Another recent BioRxiv preprint using flicker microscopy to investigate erythrocyte membrane properties is not cited, but this is a minor point as that study, while supporting some of the uninfected erythrocyte measurements and also applying flicker microscopy to rare erythrocyte variants that protect against severe malaria, does not include HbAS or HbAC erythrocytes and therefore does overlap significantly with this work. This is therefore a novel and interesting study that advances our understanding of both *P. falciparum* biology and erythrocyte membrane properties in common haemoglobinopathies.

The manuscript is well written, figures clearly presented, and arguments carefully made. However, what holds it back from being completely convincing in its current form is that critical details about the experimental design are missing, which could affect interpretation of the data, and the number of erythrocyte donors tested is on the low side. Humans and parasites are highly variable at the genotype and phenotype level. It is critical that the experiments have been repeated often enough to take that variation into account, if the authors want to make general conclusions, as they clearly do. Specific points that need addressing are as follows:

Major issues

1. Human samples

- a. Statistical robustness. The methods do not make it clear how many individuals were used to source HbAA, HbAC and HbAS erythrocytes, while the figure legends say "three to four different donors" in all cases. Firstly, it would seem essential that the specific number of independent donors be stated for each blood group/set of experiments. Secondly, 3-4 donors is a surprisingly small number to support broad claims. Ideally, additional donors should be tested; at the very least the authors should investigate whether there are systematic differences in any of the metrics between individual donors, to assess whether broad claims can indeed be made from this small number of donors. Finally, it is essential that the number of events measured should be included under the x axis in each of Figure 2-5, to allow the reader to assess robustness for themselves.
- b. Genetic confounders. HbC and HbS are not the only malaria protective polymorphisms circulating in Burkina Faso, and it is not impossible for individuals to harbour more than one polymorphism. Were donors also tested for alpha-thalassaemia and/or G6PD deficiency, to rule out individuals who had these conditions as well as HbC or HbS? Presumably compound heterozygotes for HbSC were also tested for and excluded?

2. Parasite samples

a. Synchronization. Parasites were synchronized with sorbitol treatment to “within a 6 hour window”, but how was this window actually determined? Sorbitol synchronization is often much broader than this, and can be up to 24 hours, depending on when it was done (eg. synchronization during schizont rupture and invasion is likely to produce much more tightly synchronized parasites than synchronization when the majority of parasites are already at ring stage). More specific details are required about what the time window is and how it was determined, as this will affect the heterogeneity of parasite ages used in each experiment. Given the relatively small n noted above, this could in theory lead to systematic biases between experiments unless time windows were tightly defined and made uniform between experiments.

b. Definition of stages. Erythrocytes infected with *P. falciparum* ring and trophozoite stages were compared, but both stages have quite wide time definitions - rings are generally referred to as parasites between 0-24 hours post-invasion, while trophozoites are often thought to be 24-36 hours post-invasion. It is not at all clear how the parasite ages were harmonized between experiments - only that trophozoites were enriched by magnetic column isolation. This is important - a 6 hour ring stage parasite might have very different effects on the host erythrocyte to an 18 hour ring stage parasite, and during trophozoite stages, the parasite is expanding relatively rapidly, meaning that differences in timing between may have an even bigger effect. This is similar to the issue of synchronization above - a range of different ages within a given experiment is not necessarily an issue, as long as the same range was used in all cases, and there was no systematic difference between, for example, ages used in HbAA and HbAS erythrocytes. It is impossible to determine whether such careful normalization was performed that based on the information given, and again, the small n makes this an important potential confounder.

Minor issues

1. Perhaps Figures 3 and 4 could be merged, to make it easier for the reader to compare the differential effect of HbAS and HbAC erythrocytes?
2. The knobless phenotype of the FCR3 strain variant was confirmed by “scanning electron microscopy” (line 232), but only a single infected erythrocyte is shown, which is not best practice. Was the difference actually quantified?
3. The origin of the FCR3 and knobless FCR3 strains are not stated.

Reviewer #3 (Remarks to the Author):

I was asked to just review the flickering methodology in the paper. I find it remarkably well executed and well described. A very nice application!

Responses to reviewers' comments on manuscript COMMSBIO-19-0049-T

We thank the three reviewers for their positive comments and helpful suggestions. Our detailed responses are set out below (page and line numbers refer to the "clean" manuscript).

Reviewer #1:

*General Comment: The manuscript by Froehlich et al. describes the measurements of biomechanical properties of the plasma membrane of *P. falciparum* infected red blood cells. The changes in physical parameter in the host membrane are significant because they influence pathogenicity and immune-evasion in the host. To measure the properties of the membrane the authors employed flicker spectroscopy – a technique that has not been fully explored for this particular purpose before. The advantage of the technique is that the influence of the plasma membrane and the influence of the coupling to the cytoskeleton on the overall shear modulus can be separated. Based on their measurements and values available in the literature the authors then developed a mathematical model that predicts the influence of the coupling of the membrane to the underlying cytoskeleton on the fluctuation spectrum. In order to tease out the effect of different host cell and parasite modifications the authors compared membrane properties of RBC of the haemoglobin A (WT), C and S phenotype in the infected and uninfected status as well as infected with parasites that do not produce any knobs (parasite induced RBC membrane modifications). The major findings of the paper are that *P. falciparum* infected HbAS red blood cells display a higher membrane confinement and infected HbAC red blood cells display a lower membrane confinement in comparison to infected HbAA red blood cells. At the same time this has no influence on the number and diameter of the knobs. The authors found that the different haemoglobin variants influence the bending modulus of the membrane differently and they suggest that this is due to the different oxidising states of haemichromes. These findings are novel and provide new insights into the biomechanics of mutant and *P. falciparum* infected red blood cells, however how useful they will be for a wider audience is difficult to judge. The provided measurements and the mathematical model will provide the foundation of further studies into the effect of membrane modifications on the biomechanics of red blood cells. Overall, the data presentation is clear and convincing.*

Reply: We thank the reviewer for the positive assessment of our work.

Comment 1: However, I suggest the authors also consider the following: The knobless cell line is not described in the literature reference (46); it outlines the mechanism that results in the absence of knobs (WT FCR3 was used as the knob forming variant) and prolonged culture will result in FCR3 with truncated KAHRP genes, which I assume was done. However, relevant here is that the truncation of KAHRP might result in the absence of visible knobs, but not necessarily in the absence of the complete knob structure (Rug et al., Blood 2006). This needs to be taken into consideration when discussing the results.

Reply: We thank the reviewer for this insightful comment. As suggested by the reviewer, we have fully characterized the knobless parasite line, both at the genomic and phenotypic levels. Western analysis using an antibody specific to the KAHRP protein revealed no detectable signal, and corresponding SEM images showed no knobs. Moreover, we have determined the chromosomal breakpoint by PCR using a specific *kahrp* primer and a telomeric primer and found that the breakpoint occurred within exon 1 after nucleotide 55 (amino acid 18). These data clearly show that most of the *kahrp* gene is deleted and that no remnant protein is being produced. These additional data are presented in the new supplementary Figure 3. Additionally, we removed the misleading literature reference (46) in the old manuscript, since it does not contain information on the knobless cell line investigated here.

Comment 2: Will the difference observed in the different Hb variations likely result in differences in splenic clearance and cytoadherence of infected red blood cells?

Reply: Our findings do confirm that the mechanical parameters of infected erythrocytes are heavily altered by *P. falciparum*, compared to the corresponding values obtained for uninfected erythrocytes for both HbAA and HbAS/HbAC. Although splenic clearance and cytoadhesion is beyond the scope of the manuscript, a previous study utilizing a model spleen system reported that there are no differences in the clearance rate between parasitized HbAA and HbAS erythrocytes. On the other hand, previous studies suggested that the cytoadhesion of HbAS and HbAC trophozoites is weaker than that of HbAA trophozoites.

To clarify this point, we added the following text in the revised manuscript.

“All of these changes alter the mechanical properties of the host cell membrane, which, in turn, affects the cytoadhesion and clearance mechanisms of red blood cells in the spleen. For example, previous studies demonstrated that the adhesion of HbAS and HbAC trophozoites is weaker compared to that of infected HbAA trophozoites^{1, 2}. Moreover, using a model spleen system, it has been shown that the rate of splenic clearance is > 90 % for both HbAA and HbAS trophozoites³. This finding suggests that the change in mechanical properties caused by the malaria infection is dominant, irrespective of the haemoglobin types.” Page 12, Line 308 and following.

Comment 3: The influence of lipid composition on membranes has not been considered.

Reply: As the reviewer pointed out, *P. falciparum* alters the lipid composition of the host red blood cell membrane during parasite maturation. For example, if we look at the two major phospholipids, phosphatidylcholine (PC) and phosphatidylethanolamine (PE), the percentage of PC increases during infection. Moreover, both PC and PE, exhibit a change from longer to shorter fatty acid chains, which should lead to a decrease in bending rigidity, contrary to the observed results of this study. Thus, these alterations seem to be overshadowed by the effect of parasite proteins inserted into the membrane.

Comment 4: The text and figures could be more integrated: The manuscript is based on the comparisons of the different Hb mutations/parasite strains. This is in contrast to the organization of the figures, which show different measurements of one mutation/strain. It would be more helpful to put all the measurements of the bending modulus in one figure, followed by a comparison of the surface tension, etc.

Reply: As suggested by the reviewer, we merged Figures 3 and 4 of the original manuscript and now present the bending modulus, the surface tension and the membrane confinement for HbAS and HbAC in a new Figure 3. Since no significant differences in the apparent viscosity were observed between HbAA, HbAS and HbAC, the individual determinations for the apparent viscosity for HbAS and HbAC, were moved to the supporting information section (Supplementary Figure 2).

Comment 5: On occasions it might be difficult for non-experts to follow the flow of the arguments. This can be addressed relatively easily by for example introducing the word “confinement” as “membrane confinement” in the abstract or linking conclusions more closely to the data or preceding text (e.g. line 190/line 386).

Reply: We apologize for the confusion and have amended the terminology by now using the term “membrane confinement” throughout the manuscript.

Comment 6: Line 163-165: what is the explanation that the measured values exceed the viscosity of haemoglobin solution by a factor of two?

Reply: The quantification of the internal viscosity via flicker spectroscopy has been a subject of debate for some time now. E.g., experiments on giant unilamellar vesicles with increasing internal buffer viscosities, revealed that determinations of the internal viscosity by flicker spectroscopy reproduce the correct trend, but are typically accompanied by systematic overestimations of the actual values⁴. Nevertheless, we would like to point out that the change increase in viscosity induced by *P. falciparum* as determined from the spherical harmonics approach is qualitatively correct. This point is now clarified in the discussion:

“It is widely accepted that extracting this value from flicker spectroscopy is challenging, often leading to a strong overestimation of the actual values for the cytosolic viscosity. Hence, we only regarded the qualitative changes to the apparent red blood cell viscosity during the infection with *P. falciparum*.” Page 13, Line 335 and following.

Comment 7: Line 315-317: insertion of proteins in plasma membranes does not necessarily result in the loss of membrane flexibility.

Reply: We agree with the reviewer and we have amended the text as following:

“As a result, the plasma membrane could lose flexibility. Indeed, we have measured a clear increase in the bending modulus κ (Table 1 and Fig. 2).” “Page 13, Line 320 and following”

Comment 8: Line 343: the word “extruding” for the Maurer’s cleft genesis seemed to be unfortunate.

Reply: We have replaced “extruding” by “establishing” in the main text. “Page 14, Line 349”

Comment 9: Line 371: Is there further support for the correlation that larger knob sizes influence the harmonic confinement?

Reply: Several studies have investigated the effect of knobs on the coupling between the plasma membrane and the cytoskeleton (e.g. see reference 18 in the manuscript). However, our study is, to the best of our knowledge, the first to investigate the influence of the knob size and knob density on the membrane confinement.

Reviewer #2

General Comment: This manuscript applies flicker microscopy to understand the biophysical changes that take place in erythrocytes infected with Plasmodium falciparum parasites. The parameters generated in infected and uninfected erythrocytes are compared between erythrocytes obtained from individuals who express the Haemoglobin beta variants HbS and HbC, which in heterozygotes provide protection against severe malaria. The experimentally derived values are then compared with values derived from a novel mathematical modelling approach.

Flicker microscopy has only recently been applied to understanding biophysical properties of erythrocytes, but not to my knowledge previously to P. falciparum infected erythrocytes. Previous comparable studies (which are referenced) have either only looked at uninfected erythrocytes, or uninfected erythrocytes exposed to P. falciparum adhesins. Another recent BioRxiv preprint using flicker microscopy to investigate erythrocyte membrane properties is not cited, but this is a minor point as that study, while supporting some of the uninfected erythrocyte measurements and also applying flicker microscopy to rare erythrocyte variants that protect against severe malaria, does not include HbAS or HbAC erythrocytes and

therefore does overlap significantly with this work. This is therefore a novel and interesting study that advances our understanding of both P. falciparum biology and erythrocyte membrane properties in common haemoglobinopathies.

The manuscript is well written, figures clearly presented, and arguments carefully made. However, what holds it back from being completely convincing in its current form is that critical details about the experimental design are missing, which could affect interpretation of the data, and the number of erythrocyte donors tested is on the low side. Humans and parasites are highly variable at the genotype and phenotype level. It is critical that the experiments have been repeated often enough to take that variation into account, if the authors want to make general conclusions, as they clearly do.

Reply: We thank the reviewer for his/her supportive comments.

Comment 1a: Human samples a. Statistical robustness. The methods do not make it clear how many individuals were used to source HbAA, HbAC and HbAS erythrocytes, while the figure legends say “three to four different donors” in all cases. Firstly, it would seem essential that the specific number of independent donors be stated for each blood group/set of experiments. Secondly, 3-4 donors is a surprisingly small number to support broad claims. Ideally, additional donors should be tested; at the very least the authors should investigate whether there are systematic differences in any of the metrics between individual donors, to assess whether broad claims can indeed be made from this small number of donors. Finally, it is essential that the number of events measured should be included under the x axis in each of Figure 2-5, to allow the reader to assess robustness for themselves.

Reply: We apologize for being not more specific about the number of independent determinations on which our conclusions are based. As suggested by the reviewer, we now provide detailed information regarding the number of independent determinations and the number of blood in the pertaining figures. Admittedly, the number of different donors is indeed on the lower side, yet the availability of HbAS and HbAC donors is scarce, making each blood sample precious and further experiments difficult in a reasonable time window. Irrespectively, analyzing the data derived from each donor of the same blood type separately, revealed no significant differences between the data sets (according to a one-way Anova test). This information was added to the statistical analysis section of the main text. “Page 20, Line 521 and following”

Comment 1b. Genetic confounders. HbC and HbS are not the only malaria protective polymorphisms circulating in Burkina Faso, and it is not impossible for individuals to harbour more than one polymorphism. Were donors also tested for alpha-thalassaemia and/or G6PD deficiency, to rule out individuals who had these conditions as well as HbC or HbS? Presumably compound heterozygotes for HbSC were also tested for and excluded?

Reply: We thank the reviewer for this valuable comment. All blood samples were routinely screened by HPLC in the diagnostic haematology laboratory of Heidelberg University Hospital. The phenotype was subsequently confirmed by PCR analysis. This critical information is now provided in the material and method section. “Page 16, Line 411 and following”

Comment 2a: Parasite samples a. Synchronization. Parasites were synchronized with sorbitol treatment to “within a 6 hour window”, but how was this window actually determined? Sorbitol synchronization is often much broader than this, and can be up to 24 hours, depending on when it was done (eg. synchronization during schizont rupture and invasion is likely to produce much more tightly synchronized parasites than synchronization when the majority of parasites are already at ring stage). More specific details are required about what the time window is and how it was determined, as this will affect the heterogeneity of parasite ages used in each experiment. Given the relatively small n noted above, this could in theory

lead to systematic biases between experiments unless time windows were tightly defined and made uniform between experiments.

Comment 2b Definition of stages. Erythrocytes infected with P. falciparum ring and trophozoite stages were compared, but both stages have quite wide time definitions - rings are generally referred to as parasites between 0-24 hours post-invasion, while trophozoites are often thought to be 24-36 hours post-invasion. It is not at all clear how the parasite ages were harmonized between experiments - only that trophozoites were enriched by magnetic column isolation. This is important - a 6 hour ring stage parasite might have very different effects on the host erythrocyte to an 18 hour ring stage parasite, and during trophozoite stages, the parasite is expanding relatively rapidly, meaning that differences in timing between may have an even bigger effect. This is similar to the issue of synchronization above - a range of different ages within a given experiment is not necessarily an issue, as long as the same range was used in all cases, and there was no systematic difference between, for example, ages used in HbAA and HbAS erythrocytes. It is impossible to determine whether such careful normalization was performed that based on the information given, and again, the small n makes this an important potential confounder.

Reply: We agree with the reviewers' comments 2a and 2b, and added a more detailed description of the parasite synchronization to the material and methods section of the main text. "Page 16, Line 419 and following"

Comment: 3: Perhaps Figures 3 and 4 could be merged, to make it easier for the reader to compare the differential effect of HbAS and HbAC erythrocytes?

Reply: Following the suggestions by reviewer #2, figures 3 and 4 of the original manuscript were merged, now presenting the bending modulus, the surface tension and the membrane confinement for HbAS and HbAC. Since no significant differences in the apparent viscosity were observed between HbAA, HbAS and HbAC, the individual determinations for the apparent viscosity for HbAS and HbAC, were moved to the supplementary information. (Supplementary Figure 2).

Comment 4. The knobless phenotype of the FCR3 strain variant was confirmed by "scanning electron microscopy" (line 232), but only a single infected erythrocyte is shown, which is not best practice. Was the difference actually quantified?

Reply: We now show four representative examples of the knobless phenotype of the parasite line used in our study (Supplementary Figure 3 a).

Comment 5: The origin of the FCR3 and knobless FCR3 strains are not stated.

Reply: As suggested by the reviewer, we have fully characterized the knobless parasite line, both at the genomic and phenotypic level (see also comment 1 by reviewer 1). Western analysis using an antibody specific to the KAHRP protein revealed no detectable signal and SEM showed no knobs. Moreover, we have determined the chromosomal breakpoint by PCR using a specific kahrp primer and a telomeric primer and found that the breakpoint occurred within exon 1 after nucleotide 55 (amino acid 18). These data clearly show that most of the kahrp gene is deleted and that no remnant protein is being produced. These additional data are presented in the new supplementary Figure 3.

Reviewer #3: I was asked to just review the flickering methodology in the paper. I find it remarkably well executed and well described. A very nice application!

Reply: We thank reviewer #3 for approving our methodological approach and for the kind words.

Additional changes:

Rechecking all values presented in table 1 of the main text revealed several rounding errors of the presented numbers and were corrected for accordingly. Additionally, the significance level was falsely set to $p < 0.01$ instead of $p < 0.05$, which was also corrected accordingly. These changes had no impact on the statements of this work.

1. Fairhurst RM, *et al.* Abnormal display of PfEMP-1 on erythrocytes carrying haemoglobin C may protect against malaria. *Nature* **435**, 1117-1121 (2005).
2. Cholera R, *et al.* Impaired cytoadherence of Plasmodium falciparum-infected erythrocytes containing sickle hemoglobin. *Proc Natl Acad Sci U S A* **105**, 991-996 (2008).
3. Diakit  SA, *et al.* Stage-dependent fate of Plasmodium falciparum-infected red blood cells in the spleen and sickle-cell trait-related protection against malaria. *Malaria journal* **15**, 482 (2016).
4. Betz T, Sykes C. Time resolved membrane fluctuation spectroscopy. *Soft Matter* **8**, 5317-5326 (2012).

REVIEWERS' COMMENTS:

Reviewer #1 (Remarks to the Author):

The authors have comprehensively and conclusively addressed all comments and I would like to congratulate the authors to such a nice piece of work.

Reviewer #2 (Remarks to the Author):

No further concerns - the authors have carefully and comprehensively addressed the question raised during review, and I think that the manuscript is significantly clarified and improved as a result. The author should be congratulated on a technically excellent manuscript, and also the open and proactive manner in which they have responded to the review process.